# Guided Mode Resonance Sensors with Optimized Figure of Merit

**DOI:** 10.3390/nano9060837

**Published:** 2019-06-01

**Authors:** Yi Zhou, Bowen Wang, Zhihe Guo, Xiang Wu

**Affiliations:** Key Laboratory of Micro and Nano Photonic Structures (Ministry of Education), Department of Optical Science and Engineering, Shanghai Engineering Research Center of Ultra Precision Optical Manufacturing, Fudan University, Shanghai 200433, China; 18110720008@fudan.edu.cn (Y.Z.); 16210720012@fudan.edu.cn (B.W.); 17110720004@fudan.edu.cn (Z.G.)

**Keywords:** guided mode resonance, figure of merit, biosensor, detection limits

## Abstract

The guided mode resonance (GMR) effect is widely used in biosensing due to its advantages of narrow linewidth and high efficiency. However, the optimization of a figure of merit (FOM) has not been considered for most GMR sensors. Aimed at obtaining a higher FOM of GMR sensors, we proposed an effective design method for the optimization of FOM. Combining the analytical model and numerical simulations, the FOM of “grating–waveguide” GMR sensors for the wavelength and angular shift detection schemes were investigated systematically. In contrast with previously reported values, higher FOM values were obtained using this method. For the “waveguide–grating” GMR sensors, a linear relationship between the grating period and groove depth was obtained, which leads to excellent FOM values for both the angular and wavelength resonance. Such higher performance GMR sensors will pave the way to lower detection limits in biosensing.

## 1. Introduction

Guided mode resonance (GMR) is a physical effect that occurs in thin-film structures that contain diffractive elements and a waveguide layer (“grating–waveguide” GMR structure) [1] or a planar dielectric-layer diffraction grating (“waveguide–grating” GMR structure) [2]. Light can be coupled into the waveguide modes by different grating diffraction orders. Due to periodic modulation in the waveguide, part of the guided light will leak, and thus, guided modes cannot be sustained on the waveguide grating and will interfere with the noncoupled reflected or transmitted waves [3]. The GMR effect occurs within a narrow spectral band at a particular wavelength, angle, and polarization [4].

Since Magnusson and Wang suggested the application of the GMR effect for sensing purposes due to its narrow, controllable linewidth and high efficiency [5], many researchers have shown great interest in GMR sensors, especially in biosensing [6,7,8,9,10,11,12,13]. To date, there are four main detection schemes for GMR sensors, including wavelength detection [14,15,16,17], angular shift detection [18,19,20], intensity shift detection [21,22,23,24,25], and phase shift detection [26,27,28,29]. The wavelength and angular shift detection schemes are used most often as the incident light is totally reflected with highly angular and spectral selectivity at resonance [30]. To reduce the cost of detection systems, for example, by eliminating the need for an expensive spectrometer and laser source, several intensity detection schemes have been proposed [21,22,23,24]. In addition, the sensing performance can also be enhanced by implementing phase detection schemes [26,27,28,29].

Although different detection methods result in different detection effects, the structural parameters and optical properties of GMR sensors are also important because these factors directly determine the sensors’ performance [31,32,33,34,35,36]. In a label-free biosensing system, a figure of merit (FOM), which combines the sensitivity (S) and the full width at half maximum (FWHM) of the resonance, is the accepted marker to compare the performance of biosensors, which can be defined as FOM = S/FWHM [37]. This value can simultaneously reflect the effects of the magnitude of a measured quantity change and the ability to measure small wavelength shifts [38]. In addition, a FOM can be considered as a good indicator for the expected performance of a sensor [39]. However, a narrow resonant linewidth will result in a sacrifice in the sensitivity, as resonant modes are mostly confined within the solid dielectric medium, rather than the sensing medium [40]. In contrast, highly resonant sensitivity may result in a broadened resonant linewidth, which limits the capability to measure small resonant wavelength shifts with higher accuracy [37]. To solve this problem, Ge et al. proposed an external cavity laser (ECL) GMR sensor using an active resonator laser that maintains the high resolution of laser biosensors without sacrificing sensitivity [41,42,43]. However, although the properties of sensitivity and linewidth have been investigated widely by varying structural parameters [31,32,33,34,35,36], the optimization of a FOM has not been considered for most GMR sensors. Most recently, Lan et al. investigated the FOM of a “waveguide-grating” GMR sensor by varying the geometric parameters and incident angle [44]. However, their work only investigated one GMR sensor for wavelength resonance rather than providing a design guideline that is suitable for all GMR sensors. Therefore, a convenient and effective design method to achieve an optimized FOM of GMR sensors for both angular and wavelength resonance is necessary.

In this work, by combining an analytical model and numerical simulation, we provide a convenient and efficient method as a guideline for parametric design to achieve the optimized FOM of “grating–waveguide” GMR sensors for both angular and wavelength resonance. Focusing on “waveguide–grating” GMR sensors, a linear relationship between the grating period and grating groove depth was found as a guideline to achieve optimized FOM.

## 2. Analytical Model for GMR Sensors

Lin et al. proposed a model for calculating the wavelength sensitivity of GMR sensors [35]. Briefly, as the incident light *λ* passes through the grating with an incident angle *θ_i_*, the diffraction grating equation can be expressed as follows [17]:
(1)$$\Lambda [n_{wg}\sin(\theta_{d})-n_{c}\sin(\theta_{i})]=m_{g}\lambda ,\;\;\;\;m_{g}=0,\pm 1,\pm 2,\ldots ,$$
where Λ is the grating period, *n_wg_* is the refractive index of the waveguide layer, *n_c_* is the refractive index of the surrounding medium, *θ_d_* is the diffraction angle and *m_g_* is the order of the diffracted wave. The grating diffracted wave will couple into the waveguide layer, once the diffracted wave is phase-matched to the waveguide mode, and the diffraction angle *θ_d_* is the propagation angle in the waveguide layer because the grating groove depth (*d_g_*) is assumed to be extremely thin, and thus, other influences are ignored. The guided wave condition of the planar waveguide can be defined as follows:
(2)$$k_{0}n_{wg}d_{wg}\cos(\theta_{d})-m\pi =\phi_{t}+\phi_{d},\;\;\;\;m=0,1,2,\ldots ,$$
where *k*_0_ = 2π/*λ* is the wavenumber in a vacuum, *d_wg_* is the thickness of the waveguide layer, *m* is a positive integer number that stands for the mode number of a waveguide, *φ_t_* and *φ_d_* represent phase shifts that occur due to the total internal Fresnel reflection at the waveguide grating interface and the waveguide substrate interface, respectively. For a given waveguide structure, we can use Equation (2) to calculate the *θ_d_*, and thus the incident angle *θ_i_* in Equation (1) will be obtained. When the surrounding refractive index *n_c_* is changed, the different incident angles can be solved using Equations (1) and (2), and thus the angular sensitivity *S_a_* = Δ*θ_i_*/Δ*n_c_* can be solved, where Δ*θ_i_* is the shift in the resonant angle induced by a change in the refractive index of the surrounding medium Δ*n_c_*.

As Equation (2) only stands for a single wavelength, it needs to be modified as follows to be suitable for multiple wavelengths (suitable for normal incidence only) [35]:(3)$$\frac{2\pi }{\Lambda }m_{g}d_{wg}\cot(\theta_{d})-m\pi =\phi_{t}+\phi_{d},\;\;\;\;m=0,1,2,\ldots ,$$

Using Equations (1) and (3), the wavelength sensitivity *S_w_* = Δ*λ/*Δ*n_c_* can be solved, where Δ*λ* is the shift in the resonant wavelength induced by a change in the refractive index of the surrounding medium Δ*n_c_*.

## 3. Simulation Results and Analysis

The RSoft 7.1 (RSoft Design Group, Inc., Ossining, NY, USA) based on rigorous coupled-wave analysis (RCWA) was used to simulate the GMR effect [36]. To calculate the field distributions, we used COMSOL Multiphysics 5.2a (COMSOL Inc., Stockholm, Sweden), which is based on the finite element (FEA) analysis method [45]. In our case, we simulated the GMR sensors made of a high-index material, such as silicon nitride (Si_3_N_4_, *n_g_* = *n_wg_* = 2.00) immersed in water (*n_c_* = 1.333) and with silicon dioxide (SiO_2_, *n_s_* = 1.45) as a substrate. The filling factor (FF) of the grating is fixed at 0.5 and the total thickness of the grating and waveguide layer *d* was also kept constant.

### 3.1. Angular Shift Detection Scheme for “Grating–Waveguide” GMR Sensors

For an angular interrogation technique, a monochromatic light source, such as a He–Ne laser, is typically used to measure the GMR effect in this case. The incident wavelength *λ* is set at 633 nm, and *Λ* is set at 280 nm. The total thickness of the grating and waveguide layer *d* is fixed at 100 nm. *d_g_* and *d_wg_* are the thicknesses of the grating and waveguide layer, respectively.

The schematic geometries of the angular GMR sensors are depicted in Figure 1a. The incident light can excite transverse electric (TE) or transverse magnetic (TM) modes, depending on the electric field *E_z_* or magnetic field *H_z_* perpendicular to the plane of incidence (*x*–*y* plane). When *n_c_* varies, the resonant angle *θ_i_* will change according to Equation (1). The angular position of the resonant peak will shift as illustrated in Figure 1b. Figure 1c shows the sensitivity and FWHM versus *d_g_* for TE polarization. *S_a_* increases and the resonant linewidth broadens as *d_g_* increases for TE polarization. Consequently, the angular FOM cannot achieve a higher value with higher sensitivity, as shown in Figure 1d. However, *S_a_* increases and the resonant linewidth decreases as *d_g_* decreases for TM polarization when *d_g_* is below 60 nm, as shown in Figure 1e. Figure 1f shows that, for TM polarization, the angular FOM increases as the *d_g_* decreases. A maximum FOM of 5709 was achieved when *d_g_* was 10 nm and *d_wg_* was 90 nm, which is higher than previously reported values [18]. The higher FOM not only arises from a lower *d_g_* resulting in a lower resonance linewidth, but also generates a higher sensitivity. The maximum FOM is 16.3 times higher than the minimum FOM of 350, as shown in Figure 1f.

TE and TM polarization have different performances when *d* is fixed at 100 nm. To investigate the different performances between TE and TM polarization, the COMSOL Multiphysics 5.2a (COMSOL Inc., Stockholm, Sweden) software was used to calculate the field distributions of resonant positions. Figure 1g,h, respectively, show the electric field distributions of TE polarization when *d_g_* is 50 nm and 10 nm. The electric fields are mainly stored in the waveguide layer. Therefore, a thick waveguide layer confines more electric energy, which results in a narrower resonant linewidth but reduces the sensitivity. Therefore, the value is mainly attributed to a narrow linewidth of 0.016 degrees at the cost of sensitivity, although a higher FOM of 881 was obtained for TE polarization. For the TM case, most electric energy is not confined in the waveguide layer and is distributed in the cover medium and substrate, as shown in Figure 1i,j. In the case where *d_g_* = 50 nm, up to 70.65% of the total resonance energy is distributed in the substrate, which is 82.56% of the total evanescent energy, and only 19.55% of the total resonance energy is distributed in the cover medium, which is 17.44% of the total evanescent energy. In the *d_g_* = 10 nm case, 62% and 25% of the total resonance energy is distributed in the substrate and cover medium, respectively, which respectively contain 69.72% and 30.38% of the total evanescent energy. Therefore, the evanescent energy increases by almost 13%, which leads to a higher angular sensitivity. Meanwhile, an obvious increase in the maximum electric intensity from 9.27 × 10^5^ V/m (*d_g_* = 50 nm) to 4.44 × 10^6^ V/m (*d_g_* = 10 nm) is observed for TM polarization, as shown in Figure 1i,j. This indicates that more energy was confined in the structure, thus decreasing the resonant linewidth. This result can also be proven by referencing [46], because the grating groove depth controls the coupling-loss coefficient in an exponential manner, thus the coupling-loss coefficient tends to be smaller as *d_g_* decreases. Therefore, when *d* is fixed at 100 nm for TM polarization, a shallow *d_g_* facilitates a narrow linewidth, and a relatively thick *d_wg_* (near 90 nm) also results in a higher sensitivity. A much higher FOM will be obtained thanks to the higher sensitivity and narrower linewidth.

To explain this phenomenon, we used the previous angular model to calculate *S_a_*. The red curve in Figure 2a depicts the *S_a_* of TE polarization, and a higher sensitivity region is located near *d_wg_* = 50 nm. The curved sensitivity line of TM polarization has the largest sensitivity region near *d_wg_* = 100 nm, as shown in Figure 2b. The highest sensitivity is 25.9 degrees/RIU (refractive index unit), which is very close to the highest sensitivity value in Figure 1f for TM polarization. Moreover, this largest sensitivity region *d_wg_* = 100 nm is the same as the value of the total depth of the grating and waveguide layer *d* that we set in Section 3.1. Combined with these results, the extremely high FOM accompanied by a high sensitivity and narrow linewidth shown in Figure 1f can be explained as follows: a shallower *d_g_* facilitates a narrower linewidth and a thicker *d_wg_*, approaching the highest sensitivity region (100 nm) and resulting in higher sensitivity.

To further verify the analytical result, an RCWA method was used to compare the results. In Figure 2, blue, green, and magenta marked symbols represent the numerical results for cases with a *d_g_* of 10 nm, 25 nm, and 50 nm, respectively. The results display evident differences when *d_g_* is relatively high because the influence of the grating on both the phase shift and total internal reflection cannot be ignored. However, this model is still useful for calculating the angular sensitivity when the grating depth is small. 

For optimizing the FOM of “grating–waveguide” GMR sensors, a high value region of sensitivity should be evaluated first and then *d* should be maintained near this value. Finally, by fabricating a shallower grating depth, higher sensitivity and narrower linewidth will be achieved, resulting in a higher FOM. This is a design guideline for “grating–waveguide” GMR sensors to achieve optimized FOM. 

However, the TE polarization has its highest sensitivity region near *d_wg_* = 50 nm, as shown in Figure 2a. Based on the previous deduction, we set *d* = 50 nm and simultaneously varied *d_g_* and *d_wg_* (more details are shown in Table 1). A maximum FOM (with a narrower linewidth and higher sensitivity) of 1158 was achieved when *d_g_* = 10 nm and *d_wg_* = 40 nm. The missing values in Table 1 indicate the instances when the phase-matched condition was not satisfied [17,35].

### 3.2. Wavelength Shift Detection Scheme for “Grating–Waveguide” GMR Sensors

The most common detection technique for GMR sensors is the wavelength shift-tracking method. A broadband light source is used, such as an LED light and white light source. To investigate the wavelength detection scheme, a normal incidence light was used and Λ was set at 410 nm. Similarly, we set *d* at 100 nm and *d_g_* and *d_wg_* were simultaneously varied. The schematic geometry of the GMR sensors is depicted in Figure 3a. When *n_c_* is varied, the resonant wavelength *λ* changes according to Equation (1). Thus, the resonant peak position of the wavelength will shift, as illustrated in Figure 3b.

Figure 3c,d demonstrates the relationship among resonant linewidth, sensitivity, and FOM for TE polarization. As *d_g_* decreases, a narrow linewidth and low *S_w_* is obtained, and thus the FOM cannot achieve a relatively high value combined with high sensitivity. On the other hand, for TM polarization, *S_w_* increases and the resonant linewidth decreases when *d_g_* decreases (*d_wg_* increases), as shown in Figure 3e. A maximum FOM of 2154 was achieved when the *d_g_* was 10 nm (*d_wg_* was 90 nm), as shown in Figure 3f, which is higher than those of common “grating–waveguide” GMR sensors [39].

The electric field distributions for both TE and TM polarization were simulated to further explain the previous results and are shown in Figure 3g–j. Compared to Figure 1g–j, similar results can be achieved here. For the TE mode, most of the evanescent field energy exists in the waveguide layer, and thus the sensing performances were affected. TM cases have a larger light-matter interaction region in the cover medium, and more light is strongly confined (the maximum electric intensity up to 3.61 × 10^6^ V/m), making it more suitable to achieve better performances in biosensing. The sensitivity curve of the TM polarization has the largest sensitivity region near *d_wg_* = 100 nm, which is the same as the value of *d* that we set here. On the other side, the TE polarization has the largest sensitivity region near the 50 nm of *d_wg_*, and *d* = 50 nm is maintained. Table 2 shows the difference in performance caused by varying *d_g_* and *d_wg_*. As shown in Table 2, a higher FOM is obtained with a narrower linewidth and higher sensitivity.

Combining these results with those of Section 3.1, TM polarization has better results for both angular and wavelength resonance, although different detection methods were used. This phenomenon can be explained in two respects: first, TM polarized modes have a larger phase shift length at the resonance point compared to TE modes, resulting in higher sensitivity, as shown in Figure 4. The red and blue curves represent the TM polarization phase shift in the waveguide layer at different surrounding *n_c_*. Yellow and green curves represent the TE polarization phase shift at different surrounding *n_c_*. The label *d_wg_* = 60 nm (purple curve) and *d_wg_* = 90 nm (gray curve) represent the calculated results coming from the left side of Equations (2) and (3) (called structure relation [35]). The intersection points of the structure relation curve and phase shift curve represent the points of resonant condition, where the GMR effect occurs. The length between two intersection points is proportional to resonant angle or wavelength shift. Therefore, a larger phase change length at different *n_c_*, results in greater sensitivity. In Figure 4, the TM polarization has a larger phase change length compared to the TE mode, when the *d_wg_* was 90 nm. Second, the coupling-loss coefficient for a TM polarization was smaller than that for the TE polarization, resulting in a narrow linewidth [46]. Furthermore, the phase shift length of *d_wg_* = 90 nm (purple curve) exceeds that of the *d_wg_* = 60 nm (gray curve) condition for TM polarization. This also explains why a higher sensitivity was obtained for the case where *d_wg_* = 90 nm, as shown in Figure 1f and Figure 3f.

### 3.3. Optimized FOM for “Waveguide–Grating” GMR Sensors

The schematic geometries of angular and wavelength resonance of the “waveguide–grating” GMR sensors are depicted in Figure 5a,b, respectively. Aimed at “waveguide–grating” GMR sensors, different *d_g_* values were swept by a “MOST Optimize/Scanner” (DiffractMod, RSoft 7.1, RSoft Design Group, Inc., Ossining, NY, USA) under certain *Λ* values to investigate whether a minimum resonant linewidth and a higher sensitivity exist for both the TE and TM modes. The chosen parameters are similar to those presented in Section 3.1 for the angular shift and Section 3.2 for the wavelength shift detection schemes, except *d* (*d_g_* and *d_wg_*).

For angular resonance, a better result was obtained under TM polarization; the series of results for different *d_g_* (from 150 nm to 400 nm) are shown in Figure 5c,d. Figure 5c shows the resonant linewidth and *S_a_* versus *d_g_* for TM polarization. A minimum resonant linewidth and a higher sensitivity occur for a value of *d_g_* of 300 nm. A higher FOM, almost up to 10^6^, was achieved, as shown in Figure 5d, which is 168 times greater than the highest FOM value presented in Figure 1f. For wavelength resonance, a superior result was obtained under TE polarization, and the series of results for different *d_g_* values (from 100 nm to 500 nm) are shown in Figure 5e,f. Figure 5c shows the resonant linewidth and *S_w_* versus *d_g_* for TE polarization. A minimum resonant linewidth and a higher sensitivity will occur under 380 nm of *d_g_*. Thus, a high FOM of 1618 will be achieved as shown in Figure 5d. The electric energy distribution helps us to further understand the mechanism, as shown in Figure 5a,b. For both angular and wavelength resonance, the electric field will ascend from the substrate to the grating and the surrounding medium as *d_g_* increases. Therefore, the total evanescent energy will increase in the surrounding medium, thus increasing the sensitivity. Meanwhile, an obvious increase in the maximum electric intensity from 7.44 × 10^5^ V/m (*d_g_* = 200 nm) to 3.89 × 10^7^ V/m (*d_g_* = 300 nm) was observed for angular resonance under TM polarization, as shown in Figure 4g,h. For wavelength resonance, the maximum electric intensity increased from 1.84 × 10^5^ V/m (*d_g_* = 200 nm) to 2.31 × 10^6^ V/m (*d_g_* = 380 nm) for TE polarization, as shown in Figure 5i,j. These results indicate that more energy was confined in the GMR structure, and thus, the linewidth was decreased. Combining these two points, a higher FOM value was achieved, accompanied by high sensitivity and a narrow linewidth.

Different Λ were chosen to determine the *d_g_* value that lead to a superior FOM. For wavelength resonance, different Λ varying from 330 nm to 550 nm, were chosen to calculate the position at which there is a minimum resonant linewidth and higher sensitivity for TE polarization, as shown in Table 3. The relationship among Λ, *d_g_*, and the corresponding FOM is depicted in Figure 6a.

For angular resonance, different Λ, varying from 260 nm to 320 nm, were chosen to determine the best FOM under TM polarization, as shown in Table 4. The angular sensitivity is inversely proportional to Λ. The relationship between Λ, *d_g_*, and the corresponding FOM is depicted in a three-dimensional schematic diagram in Figure 6b. All FOM values are higher than those of already reported values [18,39]. Λ as a function of *d_g_* for the wavelength and angular resonance are respectively shown in Figure 6c,d. The ratio of Λ to *d_g_* for the wavelength resonance was 0.925. A linear relationship was obtained between *Λ* and *d_g_* for the angular resonance, as shown in Figure 6d, and the formula of the best-fit line is *y* = 1097.411 − 2.808*x*. We can apply the linear relationship to design “waveguide–grating” GMR sensors with optimized FOM. 

The point of these optimized FOM results is that a narrow resonant linewidth occurred abruptly, as shown in Figure 5c,e. Norton et al. investigated linewidth of “waveguide–grating” GMR structure through coupled-mode theory [47]. Briefly, the angular and wavelength linewidth is proportional to the coupling loss coefficient, and this coefficient is determined by the overlap of the bound mode and the radiation mode. At higher grating depths, the orthogonality of the two modes decrease the overall magnitude of the coupling loss, resulting in a narrow resonant linewidth. In our work, similar resonant linewidth phenomenon can be found in Figure 5c,e. Although Norton et al. just investigated one grating period, it can be predicted that different grating periods have their own grating depths, where the narrow resonant linewidth occurs abruptly. 

## 4. Conclusions

In summary, with the aim of achieving a higher angular and wavelength FOM for GMR sensors, we systematically presented a parametric analysis elucidating the influence of structural design factors on the performance of GMR sensors. Combining an analytical model and numerical algorithm, we determine an effective and convenient method to achieve higher angular and wavelength FOM of “grating–waveguide” GMR sensors. A suitable fixed depth of the grating and waveguide will facilitate a higher FOM. This method is suitable for both angular and wavelength resonance. A linear relationship to design “waveguide–grating” GMR sensors with optimized FOMs was determined using numerical stimulations. These high FOM values can facilitate the performance of GMR sensors to achieve lower detection limits.

## Figures and Tables

**Figure 1 nanomaterials-09-00837-f001:**
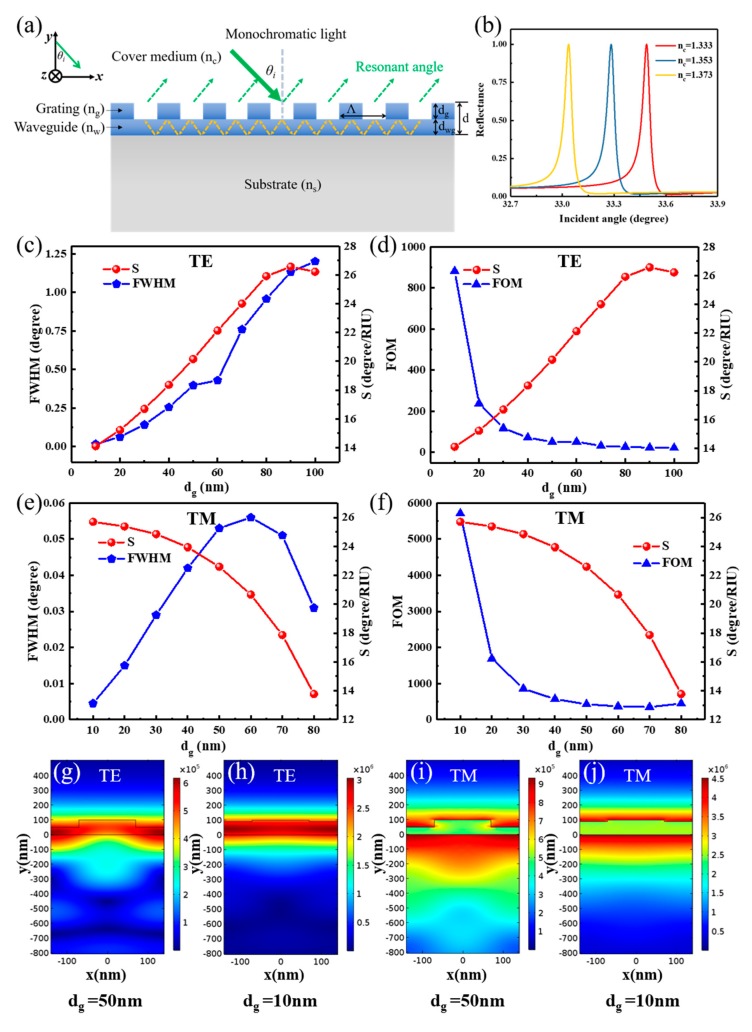
(**a**) Schematic structure of the “grating–waveguide” guided mode resonance (GMR) structure for angular resonance. (**b**) An example of reflection spectra for oblique incidence at the resonance of a monochromatic light (*d_g_* = *d_wg_* = 50 nm for transverse magnetic (TM) polarization). Calculated sensitivity, resonant linewidth and FOM versus *d_g_* for transverse electric (TE) polarization (**c**,**d**) and TM polarization (**e**,**f**). Electric field distribution of the angular resonance of the GMR structure with different *d_g_* values of 50 nm (*d_wg_* of 50 nm), 10 nm (*d_wg_* of 90 nm) and for TE (**g**,**h**) and TM (**i**,**j**). (*d* is 100 nm, Λ is 280 nm, and the filling factor (FF) is 0.5).

**Figure 2 nanomaterials-09-00837-f002:**
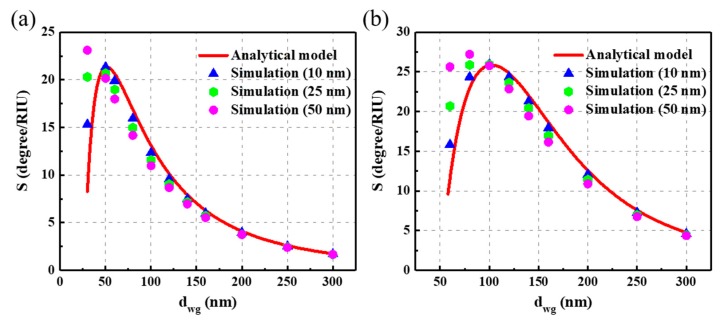
The red curved line represents the analytical model of the *S_a_* calculation for the transverse electric (TE) (**a**) and transverse magnetic (TM) (**b**) modes. Blue, green, and magenta marked symbols respectively represent calculation results of 10 nm, 25 nm and 50 nm *d_g_* of the rigorous coupled-wave analysis (RCWA) simulation method.

**Figure 3 nanomaterials-09-00837-f003:**
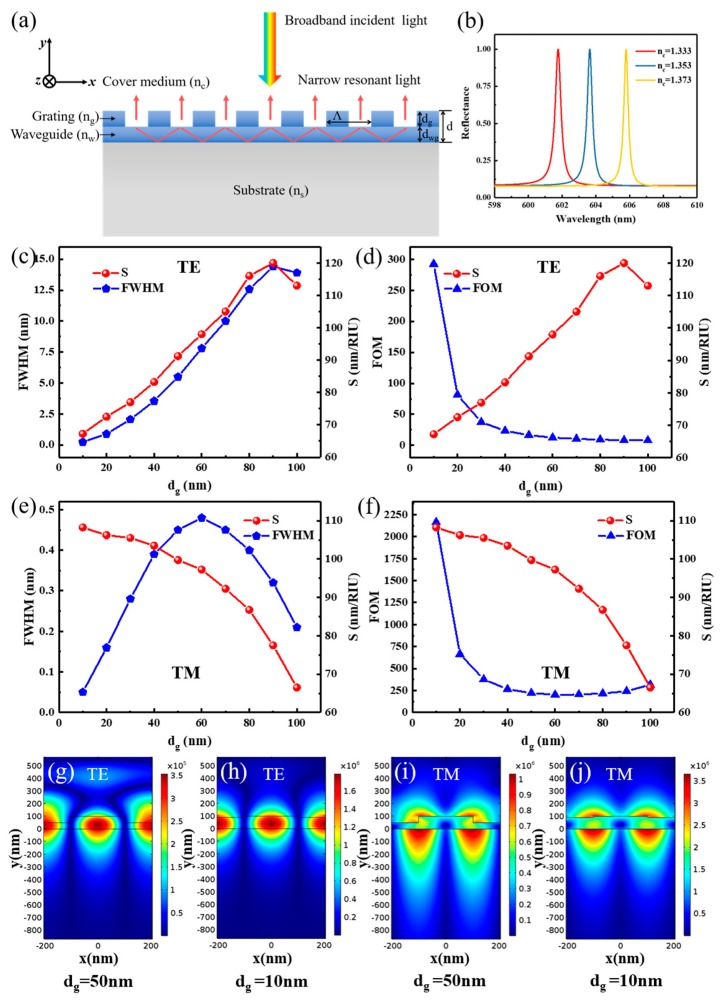
(**a**) Schematic representation of the “grating–waveguide” structure for wavelength resonance. (**b**) An example of reflection spectra for normal incidence at the resonance of a broadband light (*d_g_* = *d_wg_* = 50 nm for transverse magnetic (TM) polarization). Calculated sensitivity, resonant linewidth and figure of merit (FOM) versus dg for transverse electric (TE) polarization (**c**,**d**) and TM polarization (**e**,**f**). Electric field distribution of the wavelength resonance of the guided mode resonance (GMR) structure with different *d_g_* of 50 nm (*d_wg_* of 50 nm), 10 nm (*d_wg_* of 90 nm) and for TE (**g**,**h**) and TM (**i**,**j**), for which the resonance wavelengths are 628.91, 651.51, 601.76 and 611.26 nm, respectively. *d* is set at 100 nm, Λ is 410 nm, and filling factor (FF) is 0.5.

**Figure 4 nanomaterials-09-00837-f004:**
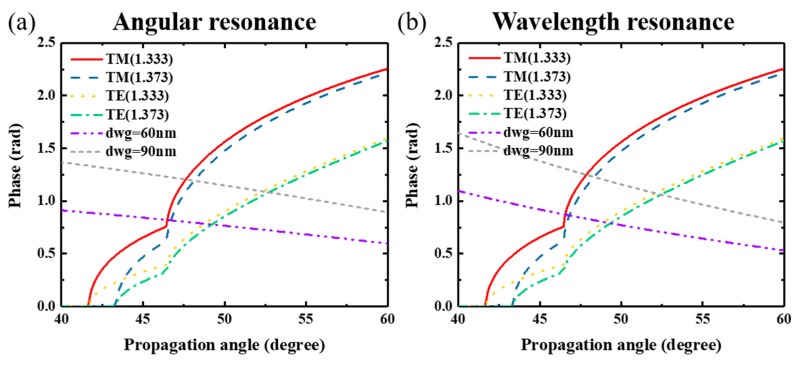
Phase shift curve in waveguide layer at different propagation angle, (**a**) angular resonance, (**b**) wavelength resonance.

**Figure 5 nanomaterials-09-00837-f005:**
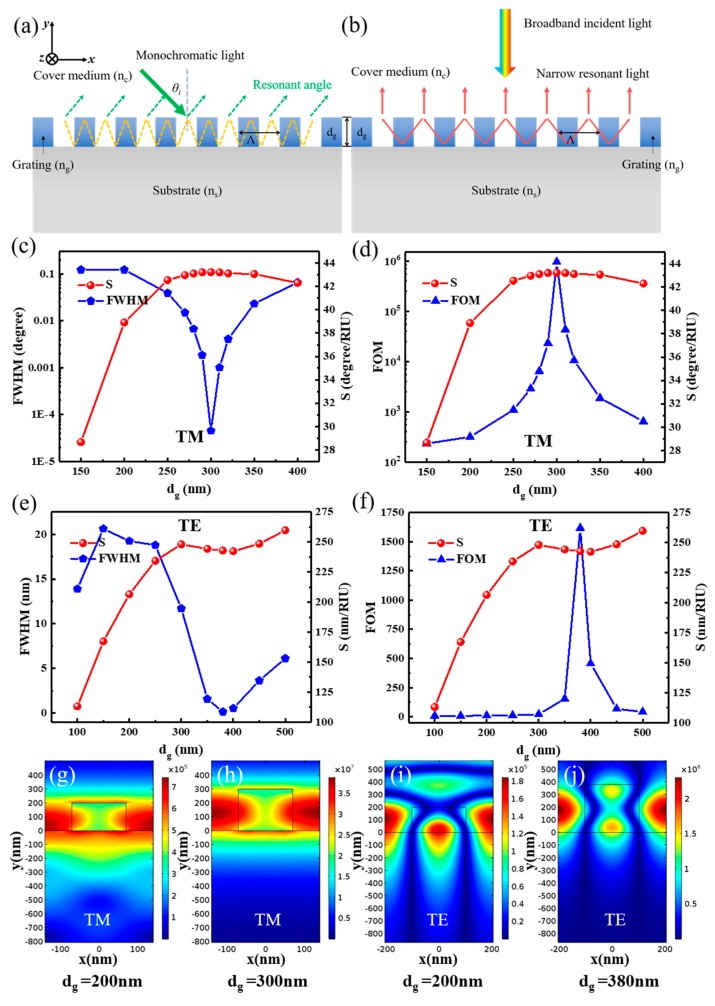
Schematic structure of the “waveguide–grating” structure for angular (**a**) and wavelength (**b**) resonance. Calculated sensitivity, resonant linewidth and figure of merit (FOM) versus *d_g_* for angular resonance under transverse magnetic (TM) polarization (**c**,**d**), and for wavelength resonance under transverse electric (TE) polarization (**e**,**f**). Electric field distribution of the “waveguide–grating” guided mode resonance (GMR) structure of the wavelength resonance with *d_g_* values of 200 nm and 300 nm for TM (**g**,**h**) and angular resonance with *d_g_* values of 200 nm, 380 nm for TE (**i**,**j**). Λ is 280 and 410 nm for the angular and wavelength resonance respectively and filling factor (FF) is 0.5.

**Figure 6 nanomaterials-09-00837-f006:**
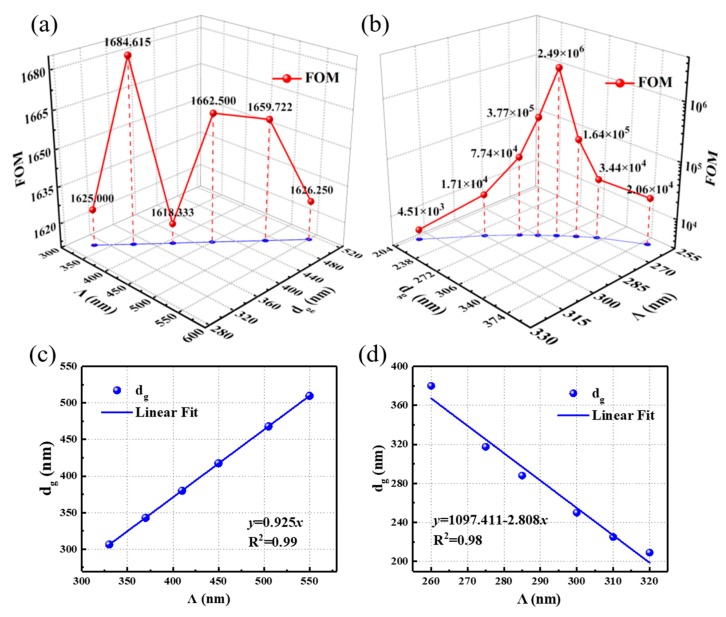
The best figure of merit (FOM) value for wavelength (**a**) and angular resonance (**b**) under different Λ and corresponding *d_g_* values. Λ as a function of *d_g_* for wavelength (**c**) and angular (**d**) resonance.

**Table 1 nanomaterials-09-00837-t001:** Linewidth (FWHM), sensitivity (S) and figure of merit (FOM) of angular resonance for *d* = 50 nm in transverse electric (TE) mode.

*d_g_* (nm)	*d_wg_* (nm)	FWHM (Degree)	*S* (Degree/RIU)	FOM
10	40	0.018	20.854	1158.567
20	30	0.058	19.138	329.959
30	20	0.083	14.934	179.923
40	10	Not a value	Not a value	Not a value

**Table 2 nanomaterials-09-00837-t002:** Linewidth (FWHM), sensitivity (S) and figure of merit (FOM) of wavelength resonance for *d* = 50 nm in transverse electric (TE) mode.

*d_g_* (nm)	*d_wg_* (nm)	FWHM (nm)	*S* (nm/RIU)	FOM
10	40	0.25	89.00	356
20	30	0.90	83.25	92.5
30	20	1.49	70.00	46.98
40	10	0.79	36.75	46.52

**Table 3 nanomaterials-09-00837-t003:** Linewidth (FWHM), sensitivity (S) and figure of merit (FOM) of the wavelength resonance under different Λ and corresponding *d_g_* for transverse electric (TE) modes.

Λ (nm)	*d_g_* (nm)	FWHM (nm)	*S* (nm/RIU)	FOM
330	306.44	0.12	195	1625
410	380.00	0.15	242.75	1618.333
450	417.50	0.16	266.00	1662.500
550	509.74	0.20	325.25	1626.250

**Table 4 nanomaterials-09-00837-t004:** Linewidth (FWHM), sensitivity (S) and figure of merit (FOM) of the angular resonance under different Λ and corresponding *d_g_* for transverse magnetic (TM) modes.

Λ (nm)	*d_g_* (nm)	FWHM (Degree)	*S* (Degree/RIU)	FOM
260	380	3.3 × 10^−3^	67.921	2.058 × 10^4^
280	302	1.7 × 10^−5^	48.322	2.486 × 10^6^
290	274	4.9 × 10^−4^	37.902	7.735 × 10^4^
310	225	3.9 × 10^−3^	30.927	7.930 × 10^3^

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
