# Peer review of "Guided Mode Resonance Sensors with Optimized Figure of Merit"

_nanomaterials, 2019, doi:10.3390/nano9060837_

Round 1
Reviewer 1 Report
Guided mode resonance sensors with optimized figure of merit
Y. Zhou, B. Wang, Z. Guo, and Z. Wu
This manuscript describes analyses of guided mode resonance sensors with a focus on optimizing the figure of merit for such devices. That is, the authors seek to identify trade-offs in design characteristics and sensitivity to optimize the figure of merit of the sensors, the ratio of the sensitivity to FWHM of the resonance.
The manuscript is interesting and potentially useful for researchers in the field. I’m pleased that the authors made the suggested changes. Overall the manuscript is improved, however there are still a few issues that should be addressed prior to publication.
1. In arriving at Eqn. 3 it appears that the authors are considering normal incidence only. This seems to be borne out in later figures but is not clear here. Is this correct?
2. Line 121/122 – The authors mention “extremely resonant linewidth” What is this supposed to mean? Perhaps you mean “extremely narrow linewidth”. What is the linewidth anyway?
3. Line 134 – The authors state “…a shallow dgfacilitates a narrow linewidth, but a relatively thick dwg(near 90 nm) also results in higher sensitivity.” This reads as if these are two separate things, but from what I can tell a shallow dgrequires dwgto be relatively thick since d is fixed; in effect, the constraint means that a relatively thick dwgfacilitates a narrow linewidth and the shallow dgresults in high sensitivity. This sentence needs to be clarified to make it clear that it’s not either/or, but one in the same.
4. For the graphs in Figs. 1, 3, and 4 make the markers for S and FWHM different. They’re indistinguishable when printed in greyscale. To the same end, avoid referring to the curves/markers by color since the reader may be looking at a greyscale printout.
5. Figure 1(i) – It’s great that the contrast has been improved on the text in this sub-figure, however the others are worse now; the text on (g), (h), and (j) are practically impossible to read, especially when printed out is greyscale.
6. Line 166 – “excellent” is a subjective term. Qualify this by making a quantitative comparison.
7. Table 1 – The last line of refers to a device that is all grating and no waveguide. What does this even mean in terms of the device that is presented in this section? How can you model a grating-waveguide without a waveguide? I would remove this from the table.
8. Line 183 – “…much better…” quantify this as this term is subjective.
9. Table 2 – The same issue as above for the 50/0 configuration. You can’t model a grating-waveguide device if you don’t have a waveguide.
Overall, the manuscript is improved and provides useful guidance on the design and optimization of waveguide and grating based resonance sensors that should be of use to researchers. This manuscript should be published and will be further enhanced by the above minor suggestions.
Author Response
Pleaser refer to the attached documents.

Reviewer 2 Report
The manuscript by Xiang et al. reports evaluations of figure of merit in guided mode resonance (GMR) sensors. The authors employed an analytical model and a finite element analysis method to explore the GMR effects in the optical sensors. By this, they systematically explored the geometrical dependence of the sensitivity and the resonant linewidth for both waveguide-grating and grating-waveguide GMR configurations. As a result, it was found that the optimal grating period is linearly correlated to the groove depth.
The results reported, in particular Figs. 5c and 5d, may be useful in developing practical GMR sensors, and thus will be good contribution to the field. However, the paper is not organized well and requires extensive revision before it can be published in Nanomaterials. I would consider recommendation of the publication after the authors addressed the following points:
1. Most of the sentences in 3.2 is quite similar to those in 3.1. Also, Figs. 1 and 3 together with the corresponding discussions are almost the same to each other. Meanwhile, it is very difficult to find any reasons to work on the two situations. The authors must first explain the point to study the angular and wavelength shifts, and then try not to repeat the same explanations/discussions on the results.
2. Section 3.3 discusses optimal sensor performance of “waveguide-grating” GMR structures. This is quite puzzling as the preceding sections are about “grating-waveguide” configurations. The authors need to make it clear why they did not examine the sensor structure optimizations of their “waveguide-grating” GMR structures and instead worked on the “grating-waveguide” GMR sensors.
3. On page 5, “a shallower d_g facilitates a narrower linewidth” of Line 151 cannot be straightforwardly assessed from their results. This is because the authors fixed d in Fig. 1. They should fix d_wg and vary dg to prove this assertion.
4. The authors need to explain why different grating periods were used in Secs. 3.1 and 3.2.
5. The manuscript only shows the simulation results without physical explanations. For instance, the linear relation in Fig. 5 better be discussed from physics points of view. Moreover, it is noted several times that “more energy was confined in the structure, thus decreasing the resonant linewidth” and “the electric field is mainly stored in the waveguide layer” without any descriptions of the underlying physics.
6. Significant figures should be corrected in Fig. 5 and Tab. 3. There are too many digits.
7. Abbreviation FEM is not for “finite element” but for “finite element analysis method”.
8. English writing needs to be improved throughout the manuscript.
Author Response

(The authors gave the same response as above.)

Round 2
Reviewer 2 Report
The authors have answered to all of the comments and revised the manuscript accordingly. I now recommend publication of this paper on Nanomaterials.
This manuscript is a resubmission of an earlier submission. The following is a list of the peer review reports and author responses from that submission.
Round 1
Reviewer 1 Report
Resonance sensors such as studied in this manuscript have true measures of sensitivity S with units nm/RIU and degrees/RIU for bulk refractive index sensing that is often of interest. The figure of merit FOM used commonly normalizes S with linewidth, spectral or angular. This is to have a measure of the sharpness of the line to monitor it with instrumentation like spectrometers without extensive curve-fitting as needed for broad lines. But the danger is that the FOM can be arbitrarily boosted by vanishing linewidths that are quite impractical in useful systems.
Many studies of sensor FOM are reported previously. A recent example is Guilian Lan, Song Zhang, Hong Zhang, Yuhang Zhu, Longyu Qing, Daimin Li, Jinpeng Nong, Wei Wang, Li Chen, Wei Wei, High-performance refractive index sensor based on guided-mode resonance in all-dielectric nano-silt array, Physics Letters A, Volume 383, Issue 13, 2019, Pages 1478-1482. The claim of this paper by Lan et al. is that FOM of 12000 is obtained; many references to FOM studies are shown there. The 12000 FOM is at 20 pm linewidth; probing this sensor would need to be done with a well collimated tunable laser—not a simple spectrum analyzer and an incoherent source.
The current work similarly has unrealistic results. In table 4 the highest FOM is at 4 microdegrees FWHM; this corresponds to around 0.1 microradian; typical lasers have divergence x1000 larger; a beam with practical divergence will not resonate here. Reasonable FWHMs are in Table 3; FOMs are at around 1600; common values appearing also in the paper by Lan et al that shows similar results; see for ex Fig 4 there.
The narrow lines with resonance of high efficiency are of course easily predicted with infinite plane wave models. A more fruitful direction would be to do the optimization within constraints dictated by realistic considerations.
In summary, there is insufficient innovative and useful content here.
Reviewer 2 Report
Manuscript No: nanomaterial 488279
Title: Guided mode resonance sensors with optimized figure of merit
Authors: Yi Zhou, Bowen Wang, Zhihe Guo and Xiang Wu1
A. Overview
1. In this manuscript the authors propose a design method for the optimization of a figure of merit for guided mode resonance sensors.
2. The contents are expressed clearly, the manuscript is well organized and is written in correct English.
3. The authors have acknowledged recent related research.
4. As long as my knowledge, the work presented is original and is correct from a scientific point of view.
B. Detailed comments
0. Abstract: is well written, objective a concise.
1. Introduction: provides interesting information and up to date references
2. Analytical model for GMR sensors: A description of the model is presented.
3. Simulation results and analysis: simulation results are presented and thoroughly discussed
C. Overall assessment
The manuscript contains novel work on design of a GMR sensor. The work reported presents reasonable utility for supplementary studies and developments in the field and it is likely to have an impact on nanomaterials readers. In my opinion, it may be published as is.
D. Review Criteria
1. Scope of Journal
Rating: Moderately high
2. Novelty and Impact
Rating: High
3. Technical Content
Rating: Medium
4. Presentation Quality
Rating: High
Reviewer 3 Report
Guided mode resonance sensors with optimized figure of merit
Y. Zhou, B. Wang, Z. Guo, and Z. Wu
This manuscript describes analyses of guided mode resonance sensors with a focus on optimizing the figure of merit for such devices. That is, the authors seek to identify trade-offs in design characteristics and sensitivity to optimize the figure of merit of the sensors, the ratio of the sensitivity to FWHM of the resonance.
The manuscript is interesting and potentially useful for researchers in the field, however there are some issues that should be addressed prior to publication.
1. Abstract – providing the FOM numbers mean very little in the abstract. In fact, the probably distract from the message because the absolute meaning of the numbers is not known at this point.
2. What is the expression for the sensitivity? Are you taking this as the partial of theta or lambda with respect to nc? This is implied but not necessarily clear.
3. Figure 1(i) – the text on the inset is practically impossible to read, especially when printed out is greyscale.
4. Just a couple of typos and formatting issues with the equations that should be corrected in the final editing stage.
Overall, the manuscript provides some useful information on the design and optimization of waveguide and grating based resonance sensors that should be of use to researchers. For example, “a high value region of sensitivity should be evaluated first and then d should be maintained near this value” provides the reader with the rules for optimizing the grating-waveguide structure. This manuscript should be published and might be enhanced some by the above minor suggestions.
